# Characterization of Atypical Protein Tyrosine Kinase (PTK) Genes and Their Role in Abiotic Stress Response in Rice

**DOI:** 10.3390/plants9050664

**Published:** 2020-05-23

**Authors:** Allimuthu Elangovan, Monika Dalal, Gopinathan Kumar Krishna, Sellathdurai Devika, Ranjeet Ranjan Kumar, Lekshmy Sathee, Viswanathan Chinnusamy

**Affiliations:** 1Division of Plant Physiology, ICAR-Indian Agricultural Research Institute, New Delhi 110012, India; elangovan_11098@iari.res.in (A.E.); krishna.kumar@nmims.edu (G.K.K.); devika_11100@iari.res.in (S.D.); lekshmy.s@icar.gov.in (L.S.); 2ICAR-National Institute for Plant Biotechnology, New Delhi 110012, India; 3Division of Biochemistry, ICAR-Indian Agricultural Research Institute, New Delhi 110012, India; ranjeet_biochem@iari.res.in

**Keywords:** BRI1, drought, dual specificity, osmotic stress, receptor-like protein kinases

## Abstract

Tyrosine phosphorylation constitutes up to 5% of the total phophoproteome. However, only limited studies are available on protein tyrosine kinases (PTKs) that catalyze protein tyrosine phosphorylation in plants. In this study, domain analysis of the 27 annotated *PTK* genes in rice genome led to the identification of 18 PTKs with tyrosine kinase domain. The kinase domain of rice PTKs shared high homology with that of dual specificity kinase BRASSINOSTEROID-INSENSITIVE 1 (BRI1) of *Arabidopsis*. In phylogenetic analysis, rice PTKs clustered with receptor-like cytoplasmic kinases-VII (RLCKs-VII) of *Arabidopsis*. mRNAseq analysis using Genevestigator revealed that rice *PTKs* except *PTK9* and *PTK16* express at moderate to high level in most tissues. *PTK16* expression was highly abundant in panicle at flowering stage. mRNAseq data analysis led to the identification of drought, heat, salt, and submergence stress regulated *PTK* genes in rice. *PTK14* was upregulated under all stresses. qRT-PCR analysis also showed that all *PTKs* except *PTK10* were significantly upregulated in root under osmotic stress. Tissue specificity and abiotic stress mediated differential regulation of *PTKs* suggest their potential role in development and stress response of rice. The candidate dual specificity *PTKs* identified in this study paves way for molecular analysis of tyrosine phosphorylation in rice.

## 1. Introduction

Protein kinases catalyze phosphorylation of serine (Ser), threonine (Thr), and tyrosine (Tyr) residue of target proteins by transferring the γ-phosphate from ATP. Based on their substrate specificity to phosphorylate Ser/Thr and Tyr residues, protein kinases were initially classified into Ser/Thr kinases and Tyr kinases, respectively [1]. Later dual-specificity protein tyrosine kinases (DsPTKs) which phosphorylate all three residues viz. Ser, Thr, and Tyr were also identified [2]. Among these, serine phosphorylation (pS) is most common followed by threonine (pT) and tyrosine phosphorylation (pY). In plants such as rice and *Arabidopsis*, the relative abundance of pS and pT is about 80–85% and 10–15%, respectively, while pY ranges between 0–5% [3,4,5]. The lower representation of pY in phosphoproteome has been attributed to occurrence of Tyr phosphorylation on less abundant proteins, methods of detection and sample type, i.e., subcellular fractions or whole-cell extracts [3,6]. In spite of lower abundance, pY has been shown to plays an important role in cell division, growth, and differentiation in humans. Impairment of PTK activity has been associated with diabetes, while hyperactivity leads to oncogenesis [7]. In plants, Ser/Thr kinases are the most studied kinases, while Tyrosine kinases are relatively less known. Tyr phosphorylation was first detected in *Pisum sativum* by immunoblotting with anti-phospho-Tyr antibodies [8]. Later pharmacological studies using inhibitors demonstrated the role of Tyr phosphorylation in different plant processes, for example, organization of microtubules [9], Abscisic acid (ABA) pathway in seeds [10], and leaf bending in *Mimosa pudica* L. [11]. The involvement of Tyr phosphorylation in different stages of plant development starting from embryo to leaf senescence and abiotic stress response reiterates its significance in plants [12,13,14,15]. Furthermore, inhibitors of protein tyrosine phosphatases (PTPs) have been shown to inhibit ABA-mediated stomatal closure [16,17]. A loss-of function mutation in a dual-specificity PTP (DsPTP) *PHS1* conferred hypersensitivity to the plant stress hormone ABA and thus enhanced expression of ABA-dependent RAB18 gene [18]. Recently, poplar PTP1 was found to be a negative regulation of salt tolerance [19]. These evidences suggest that tyrosine phosphorylation and dephosphorylation play important role in stress response of plants.

Structurally, protein kinases share a conserved catalytic domain of ~250 to 300 amino acids with eleven conserved sub-domains (I to XI). A conserved lysine residue in the subdomain II helps in binding to ATP, and is essential for maximal enzyme activity. The presence of aspartic acid residue in the subdomain VII is important for the kinase activity and subdomain VI helps in recognition of specific hydroxyl amino acid [20,21]. The presence of DLRAAN or DLAARN in sub-domain VI indicates Tyr specificity, while the presence of DLKPEN indicates Ser/Thr specificity. Subdomain VIII helps in recognition of peptide substrates, where, Tyr specificity is defined by the highly conserved consensus sequence PI/VK/RWT/MAPE and the relatively less conserved GT/SXXY/FXAPE sequence confers Ser/Thr specificity. The subdomain XI consensus motif CW(X)6RPXF is highly specific to tyrosine kinase and it was found to be conserved in tyrosine kinases from mammals, fruit fly, and Dictyostelium [21]. Hence, based on CW(X)_6_RPXF motif, 57 PTKs were identified in *Arabidopsis*. However, all these *Arabidopsis* PTKs have KXXN in sub-domain VIB, which is a Ser/Thr specific motif. Based on this observation, it was inferred that plants do not encode bona fide PTKs, and hence these kinases were inferred as dual specificity protein kinases (DsPTKs) [21].

Though typical PTKs were predicted in rice, yet the number was far less (3 in *O. sativa* ssp. *indica* and four in *O. sativa* ssp. *japonica*) [22]. Moreover these were predicted based on the residues known to be essential for catalytic activity, hence their role as typical tyrosine kinases still remains to be verified. Therefore, to characterize the PTKs in rice genome, we carried out the in silico analysis of annotated *PTK* genes in rice genome and their expression analysis under osmotic and drought stress at seedling and reproductive stages respectively.

## 2. Results and Discussion

### 2.1. Analysis of PTK Genes in Rice

Twenty seven genes annotated as “tyrosine protein kinase domain containing protein” were identified from rice genome RGAP Release 7. Based on pfam domain analysis, 19 of these 27 annotated proteins were identified with protein tyrosine kinase domain (PF07714.16). Further, HMMER analysis also identified 19 proteins with protein tyrosine kinase domain (PF07714.16) and remaining 8 with protein kinase domain (PF00069.24) based on the lowest E-value. Hence, these 19 putative PTKs were further analyzed for presence of conserved motifs in 11 (I-XI) subdomains characteristic to protein kinases. The subdomains for ATP binding and catalysis were conserved among the 19 putative PTKs of rice. The subdomain I of protein kinases is characterized by GxGxϕGXV, where ϕ can be F or Y. Among the putative PTK proteins, LOC_Os02g39560 lacked sub domain I as well as many of the conserved residues in other sub domains. Hence it was not considered for further analysis. The remaining 18 candidate PTK proteins of rice showed highly conserved GxGxFGXV motif in subdomain I, invariant Lysine (K) residue in subdomain II, Glutamic acid (E) in subdomain III and motif DFG in subdomain VII which are required for anchoring of ATP and enzyme catalysis, respectively [23,24].

Protein kinases are differentiated as Ser/Thr or Tyr protein kinases based on subdomain VI, VIII, and XI [16,18]. The tyrosine kinases are characterized by presence of consensus sequence of DL(R/A)A(A/R)N, XP(I/V)(K/R)W(T/M)APE, and CW(X)6RPXF in subdomains VI, VIII, and XI, respectively. Of the 18 candidate PTK proteins of rice, 17 showed motifs (Y/H)RDFK(A/T)SN, TXXYXAPE, and CL(X)_6_RPXM at subdomain VI, VIII and XI, respectively (Figure 1) instead of typical tyrosine kinase specific motifs found in animal system viz., HRDLXARN, PXXWXAPE, and CW(X)_6_RPXF [24]. Of the 18 candidate PTK loci identified from rice, LOC_Os07g31290 has all subdomains except subdomain XI. Since LOC_Os07g31290 showed expression in both in silico and qRT-PCR analysis, it was also included in this analysis. Notably the consensus sequence for subdomain XI showed a conserved motif CL(X)_6_RPxM among the rest of the 17 candidate PTK protein sequences. These results prompted us to analyze already validated dual specificity protein kinases from *Arabidopsis*.

In *Arabidopsis*, a classical example of tyrosine phosphorylation is presented by brassinosteroid receptor BRASSINOSTEROID-INSENSITIVE 1 (BRI1) and its co-receptor BRI1-ASSOCIATED KINASE 1 (BAK1). BRI1 and BAK1 are leucine-rich repeat receptor-like kinases (LRR-RLKs) which were initially classified as serine/threonine protein kinases. Later it was found that both the cytoplasmic domains of BRI1 and BAK1 also autophosphorylate on tyrosine residues, thereby proved that these two are dual-specificity kinases [25]. Sequence analysis revealed that these two kinases do not have the typical tyrosine kinase specific CW(X)_6_RPXF motif of subdomain XI reported for animal PTKs. Furthermore, sequence alignment of BRI1 with the 18 PTK proteins identified in rice revealed >80% identity in consensus sequence of subdomains VI, VIII, and XI (Figure 2). This indicates that the 18 PTKs identified in rice might function as dual specificity BRI-like kinases. There is a high probability that in the absence of bona fide tyrosine kinases, these members of kinase family might be carrying out tyrosine phosphorylation. This is further supported by fact that members of the plant calcium-dependent protein kinase (CDPK/CPK)-related PKs (CRK) superfamily, in addition to Ser/Thr phosphorylation, can also catalyze trans- and autophosphorylation of Tyr residues in *Arabidopsis* [26]. The identification and characterization of kinases capable of phosphorylating tyrosine residues would be an important aspect of understanding the signaling pathways in plants.

We named the rice candidate PTKs identified in this study as PTK1 to PTK18 based on their chromosomal localization starting from chromosome 1 to 12 in an ascending order (Appendix A). These *PTK* genes were distributed across nine chromosomes with 1 to 3 locus per chromosome except chromosome 5, 11, and 12 which do not harbour any *PTK* gene (Appendix A). Gene structure analysis revealed that the *PTK* genes have 1–6 introns (Figure 3).

The protein length of PTKs varied from 376 to 500 amino acids, while the molecular weight ranged from 41.43 to 55.05 kDa. The pI (isoelectric point) of the PTKs varied from 5.69 to 10.24 (Appendix A). Based on TargetP 1.1 prediction, there were four PTKs viz. PTK11, PTK14, PTK15, and PTK17, with high probability of chloroplast localization, PTK12 contained a signal peptide for secretory pathway, while rest of the PTKs were predicted to be localized in other locations. LOCALIZER1.0 predicted nuclear localization signal (NLS) in PTK1, PTK2, PTK5, PTK7, PTK8, PTK9, PTK10, PTK13, and PTK17, while NLS, cTP (chloroplast transit peptide) and mTP (mitochondrial transit peptide) in PTK12, NLS and cTP in PTK14, and mTP and NLS in PTK15 (Appendix A).

To understand the phylogenetic relationship with *Arabidopsis* PTKs and to identify the subfamily of kinases, rice PTK sequences were used as query and 18 *Arabidopsis* homologs were identified. The analysis of homologs in the phylogenetic tree revealed that most of the *Arabidopsis* homologs of rice PTKs belonged to group VII of receptor-like cytoplasmic kinases (RLCKs), which are mainly AVRPPHB SUSCEPTIBLE 1 (PBS1)-like 1 (PBL) proteins (Figure 4). The orthologs of *OsPTK1*, *OsPTK4* and *OsPTK15* in *Arabidopsis* encode PBL proteins. The PBL proteins are involved in plant innate immunity [27,28,29,30]. Besides their role in innate immunity, *OsPTK1* orthologs of Arabidopsis *PBL34* (AT3G01300) and *PBL35* (AT3G01300) have also been shown to regulate vitamin E biosynthesis in seeds [31]. Orthologs of *OsPTK6* and *OsPTK12* in Arabidopsis *COLD-RESPONSIVE PROTEIN KINASE 1* (*CRPK1*, AT1G16670) [32] and *Calcium/Calmodulin-Regulated Receptor-Like Kinase 1* (*CRLK1*, AT5G54590) [33], respectively, have been shown to regulate cold tolerance. RLKs contain a ligand binding ectodomain, a transmembrane domain, and a cytoplasmic kinase domain [34]. RLCKs are a subfamily of RLKs which lack transmembrane domains and ligand binding domain but retain the intracellular domains with kinase activity. There are about 162, 160, and 402 *RLCK* genes in maize, rice, and *Arabidopsis*, respectively [35,36,37]. Based on phylogenetic analysis these RLCKs have been divided into 15–17 subgroups in *Arabidopsis*, rice and maize [34,37]. However, these RLCKs were not further classified based on substrate specificity.

To understand the regulation of *PTK* genes, in silico analysis of 2kb promoter region of *PTK* genes was carried out. The analysis revealed presence of several elements such as light responsive element, wound/elicitor response elements, abiotic stress response elements related to drought and salt stresses, and hormone responsive element associated with plant hormones such as ABA, IAA, GA and Ethylene (Appendix A). Among *cis*-elements there was high propensity of recognition site for dehydration, salt, pathogenesis and wound response. *Cis*-elements related to root, mesophyl cell, grain and pollen specific expression were also highly presented in these promoters.

### 2.2. mRNAseq Expression Analysis of PTK Genes

The expression potential of *PTKs* in different tissues, developmental stages and abiotic stresses was analyzed by using GENEVESTIGATOR. The cluster analysis based on the expression levels in different tissues and developmental stages revealed that expression levels of *PTK16* and *PTK9* were very low except at flowering stage where *PTK16* showed high level of expression in panicles (Figure 5). Most of the *PTK* genes were highly expressed in root. The *cis*-regulatory element (CRE) *ROOTMOTIFTAPOX1*, which is known to induce high expression in roots [38,39], was found in all *PTKs* (Appendix A).

*PTK1* was ubiquitously expressed in all the tissues with high levels of expression at booting and heading stage in spikelets, pistil, and anther. *PTK3*, *PTK4*, *PTK7*, *PTK8*, *PTK10*, *PTK17*, and *PTK18* were highly expressed in vegetative (coleoptiles, root/root tips, shoot/leaf, flag leaf) and reproductive tissues (panicle), and hence expressed during all developmental stages of rice. *PTK1*, *PTK6*, *PTK13*, *PTK14*, and *PTK15* showed high level of expression in anther and pistil but their expression levels were lower in leaf/shoot and flag leaf. Expression levels of *PTK2*, *PTK5,* and *PTK11* were highest in root followed by leaf (Figure 5). In consistent with expression of *PTKs* at flowering stage in anthers, the CREs *POLLEN1LELAT52* and *GTGANTG10* [40], which regulate expression in pollens in rice, were found to be highly represented in the promoter of all the *PTKs*. At milk stage of grain development all *PTKs* were downregulated except *PTK1*, *PTK3*, and *PTK10*. Interestingly, during dough stage and mature grain stage, *PTK4*, *PTK7*, *PTK10*, and *PTK18* were upregulated which suggests a role for these genes in grain maturation. *PTK12*, *PTK13*, and *PTK14* were highly expressed up to flowering stage but were downregulated during grain development (Figure 5b). The overrepresentation of CRE *EBOXBNNAPA* for seed specific expression [41] was found in promoter of *PTKs* (Appendix A).

Stress responsive expression pattern of *PTKs* were analyzed using mRNAseq data from different abiotic stress experiments. The analysis revealed that *PTK11* and *PTK16* were upregulated in root tissues on 2nd and 3rd day, while *PTK14* was upregulated only on 2nd day of drought stress. Expression levels of *PTK2*, *PTK4*, *PTK6*, *PTK8*, *PTK13*, *PTK17,* and *PTK18* were significantly downregulated in root on 2nd and 3rd days of drought stress (OS-00230; Figure 6). *PTK2* and *PTK3* showed 1.5 to 2 folds increase in leaf tissues after 13 to 26 days of drought stress (OS-00369; Figure 6).

In another drought stress experiment (OS-00143), drought tolerant (DT) donor parent PSBRC28 (P28), DT introgression line H471 and drought sensitive (DS) recurrent parent Huang-Hua-Zhan (HHZ) were grown in control (20% soil moisture) and drought stress (soil moisture 15%, 10%) at tillering stage [42]. *PTK3*, *PTK7*, and *PTK14* showed upregulation in leaf tissues at least in one genotype under drought stress. It is interesting to note that *PTK3* was upregulated in leaves at 15% not at 10% AWC, while *PTK7* was upregulated in leaves at 10% not at 15% AWC in all three different genotypes. *PTK4*, *PTK5*, *PTK11*, *PTK13*, and *PTK17* showed significant downregulation under drought stress in leaf tissues in all three genotypes (OS-00143; Figure 6). *PTKs* were found to be upregulated in seeds under heat stress with 1.5 to 2.5 fold increase in expression as compared with control except *PTK3*, *PTK9*, and *PTK12* (OS-00155; Figure 6). Expression levels of *PTKs* were largely unaltered under salt stress except *PTK14* which showed about 1.5 fold increase at 24 h of salt stress (OS-00148; Figure 6).

The anther transcriptome analysis of photo thermo sensitive genic male sterile lines (PTGMS) rice Y58S and Pei’ai64S (P64S) under cold stress [43] revealed that 1.25 to 2 fold upregulation of *PTK2*, *PTK5*, *PTK8*, *PTK14*, *PTK16*, and *PTK18* in P64S, a genotype relatively tolerant to cold, while these genes were downregulated by 1.5 to 2 fold in cold susceptible Y58S. Only *PTK13* showed 1.5 fold increase in expression levels in both genotypes (OS-00153; Figure 6). Interestingly *PTK2*, *PTK5* and *PTK6*, *PTK14*, *PTK16*, and *PTK18* showed upregulation in relatively cold tolerant genotype but were downregulated in susceptible genotype. This suggests their potential role in spikelet fertility under cold stress.

In submergence stress experiment with deepwater rice cv. C9285 and non-deepwater rice cv. Taichung 65 (T65) [44], several *PTKs* were upregulated (OS-00362; Figure 6). *PTK6*, *PTK13* and *PTK14* appears to be early submergence regulated genes as these were upregulated even at 1 h of submergence stress in deepwater rice cv. C9285, while only *PTK13* showed about 1.5 fold increase in expression in T65. *PTK6* expression was upregulated throughout the stress period in deepwater rice cv. C9285. *PTK8* and *PTK18* showed upregulation in expression (1.5 to 2 fold) even at 12-24 h in both deepwater rice and non-deepwater rice genotypes. *PTK1*, *PTK2*, and *PTK4* were preferentially upregulated in non-deepwater rice T65, while *PTK6* and *PTK14* were upregulated in deepwater rice cv. C9285. Interestingly *PTK14* showed highest increase in expression after 1 h submergence in C9285, while it was significantly downregulated in non-deepwater rice cv. T65 (Figure 6). *PTK8*, *PTK13*, and *PTK18* may be involved in general submergence tolerance responses as these were upregulated in both tolerant and susceptible genotypes, while *PTK14* appears to be associated with specific mechanism of submergence tolerance in deep-water rice.

### 2.3. Real-Time RT-PCR Analysis of Expression of PTK Genes

The stress responsive expression of *PTK* genes was further analyzed by quantitative RT-PCR analysis at seedling stage and anthesis stage in drought tolerance rice cv. Sahbhagi Dhan (IR74371-70-1-1). It is a drought tolerant rice cultivar with wider adaptability. It was also released in Nepal and Bangladesh in the name of Sukha Dhan 3, and BRRI Dhan 56, respectively. At seedling stage, osmotic stress was imposed for two weeks. *PTK1* and *PTK10* were upregulated by 2 and 2.5 folds in shoots at seedling stage under osmotic stress (Figure 7).

In contrast, all the *PTKs* except *PTK10* were upregulated in roots at seedling stage under osmotic stress. The level of expression in root tissues was in the range of 2.6 to 6.5 fold except *PTK14* with the highest (46.75 folds) induction under osmotic stress in the drought tolerant rice cultivar Sahbhagi Dhan (Figure 7). Although most *PTK*s showed high level of expression in root tissues during normal development, only *PTK11*, *PTK14*, and *PTK16* were found to be upregulated under drought stress in roots in Genevestigator datasets (Figure 5 and Figure 6).

*PTK4* showed significant upregulation (6 fold) in the roots of drought tolerant rice cv. Sahbhagi Dhan in our study (Figure 7). In an earlier study, *bbs1* (*bilateral blade senescence 1*) rice mutant with early leaf senescence character was identified from EMS mutagenesis. Map-based cloning revealed that *bbs1* is a single base insertion in LOC_Os03g24930 (i.e., *PTK4* in our study) which led to loss-of function. *BBS1* expression was upregulated up to 8 h and later down regulated under salt stress. The *bbs1* mutant also showed hypersensitivity to salt stress [45]. In our study *PTK4/BBS1* expression was upregulated under osmotic stress. Our results suggest that in addition to salt stress [45], *PTK4/BBS1* may also confer osmotic stress tolerance to rice.

To analyze the expression of *PTK* genes at anthesis stage, drought stress was imposed to the field grown plants of rice cv. Sahbhagi Dhan at anthesis stage by withholding irrigation. Flag leaf and panicle tissues were sampled for gene expression analysis when the soil moisture decreased to about 6% in the top 15 cm soil layer (Appendix A) and the leaf relative water content was 64.3 ± 3.1% in the drought stressed plants (Appendix A). None of the *PTKs* were upregulated in flag-leaf and panicle. *PTK6* was downregulated in flag leaf, while in panicle *PTK4* and *PTK18* were downregulated under drought stress.

Although osmotic stress upregulated most of the *PTKs* in roots, only 1–2 *PTKs* were upregulated in leaf tissues under osmotic stress caused by PEG, drought and salt. In contrast, under temperature (cold and heat) stresses, many PTKs were upregulated (Figure 6 and Figure 7). The reasons for upregulation of fewer genes under drought and salt stress as compared to that under temperature stresses may due to the target tissues and genotypes analyzed. For example, drought and salt stress studies were done in vegetative tissues (root and leaf), while the temperature stress studies were done in reproductive tissues (anther and seed).

As ABA-dependent signaling plays a major role in stress inducible gene expression [46], mining of microarray data from RiceExPro was carried out. The results showed that ABA-mediated expression of *PTKs* in roots of rice seedlings is highly dynamic over durations of the treatment. ABA treatment moderately upregulated *PTK3* and *PTK13* at 1 h, but were downregulated at 6 h in roots. *PTK2*, *PTK11*, *PTK14*, *PTK15*, and *PTK16* were upregulated by ABA at 3–6 h, especially *PTK11*, while *PTK5* and *PTK12* were also upregulated at 6h in roots of rice seedlings. However, *PTK7*, *PTK8*, and *PTK18* were downregulated by ABA at 3–6 h in roots (Figure 8). In the shoots of rice seedlings, *PTK11* was highly upregulated at 6 h of ABA treatment, while *PTK5*, *PTK6*, and *PTK13* were moderately upregulated (Figure 8). Hence these 13 *PTK* genes may be regulated through ABA-dependent pathway under stress. Of these ABA-regulated *PTKs*, *PTK6*, *PTK11*, and *PTK13* were regulated in both root and shoot, and rest 10 *PTKs* were regulated only in root.

Jasmonic acid (JA) cross-talks with ABA and regulate abiotic stress tolerance. Some of the *OsPTK* orthologs in *Arabidopsis* have been shown to regulate plant immunity against hemibiotrophic pathogens [30,47]. Hence, we examined the expression of *OsPTKs* to understand whether they are also regulated by JA-dependent pathway. *PTK5*, *PTK8*, *PTK11*, *PTK13*, and *PTK15* were upregulated at 1 h in roots by JA, but *PTK13* and *PTK15* were downregulated at 3–6 h. By 3 or 6 h of treatment, *PTK3*, *PTK5*, *PTK7*, *PTK11*, *PTK17*, and *PTK18* were also upregulated in roots by JA (Figure 8). As regards shoots, PTK11, PTK13, and PTK17 were upregulated by JA at 3 and/or 6 h of treatment (Figure 8). Interestingly, *PTK11* and to a lesser extent, *PTK5* and *PTK13* were upregulated both in roots and shoots by both ABA and JA (Figure 8).

All the *PTKs* examined showed upregulation in at least one of abiotic stress condition. *PTK2*, *PTK8*, *PTK13*, and *PTK18* were upregulated under osmotic, heat, cold and submergence stresses, while *PTK14* was upregulated under all abiotic stresses examined (Figure 6 and Figure 7). Promoter analysis also revealed abundance of stress and ABA regulated CREs such as *MYBCORE* and *MYCCONSENSUSAT* in *PTKs* (Appendix A).

Some of the *Arabidopsis* homologs of rice PTKs have been shown to play important roles in biotic and abiotic stress tolerance, and development. For example, At4G35600 (CAST AWAY), a homolog of *OsPTK11*, has been shown to inhibit organ abscission in *Arabidopsis* [48]. *At1G61590* (*SCHENGEN1*), an ortholog of *OsPTK4*, is required for the positioning and correct formation of the centrally located Casparian strip membrane domain proteins (CASPs) domain endodermal cells in *Arabidopsis* [49]. In our study *PTK4* was specifically found to be upregulated in root under drought (Figure 6). Casparian strip formation is an important adaptive trait for drought tolerance. Hence, it will be worth examining the role of this kinase in Casparian strip formation in rice under drought. AT5G54590, an ortholog of *OsPTK12*, codes for *CALCIUM/CALMODULIN-REGULATED RECEPTOR-LIKE KINASE 1* (*CRLK1*), plays important role in cold tolerance of *Arabidopsis* [50]. At2G05940 (*RIPK/ACIK1A/PBL14*) and At5G35580 (*AvrPphB SUSCEPTIBLE1-LIKE13, PBL13*), homologs of *OsPTK4* and, At2G02800 (PBL3) and At5G01020 (*PBL8*), homologs of *OsPTK11* and *OsPTK18*, respectively, have been shown to confer immunity against pathogenic bacteria in *Arabidopsis* [27,51]. *AtPBL13* knockout mutation resulted in enhanced reactive oxygen species (ROS) burst in response to bacterial attack [27]. Since ROS signaling and ROS management are key for abiotic stress tolerance, and abiotic stresses differentially regulate the expression of *OsPTK4*, *OsPTK11* and *OsPTK18* in rice (Figure 6), it is tempting to speculate the that these PTKs may play a role in abiotic stress tolerance through ROS signaling and management.

## 3. Materials and Methods

### 3.1. Plant Growth Conditions and Treatments

Rice (*O. sativa* ssp. *indica*) variety Sahbhagi Dhan (IR74371-70-1-1) was used for expression analysis in the present study. Two experiments were conducted one each at seedling and reproductive stage. In experiment with seedlings, seeds were germinated on germination paper under culture room conditions. Five-day old uniform rice seedlings were transferred to Yoshida solution. For imposing osmotic stress, Yoshida’s solution was supplemented with PEG6000 i.e., 100 g PEG6000 was dissolved in 1000 mL Yoshida’s solution. Seedlings grown in Yoshida solution without PEG served as control. The seedlings were grown for 21 days in hydroponics at 25 °C under 16 h/8 h light/dark conditions.

For imposition of drought stress at reproductive stage, 21 days old seedling were transplanted in 1 m^2^ puddled plots and grown under natural field conditions. Drought stress was imposed by with-holding irrigation at booting stage till soil matric potential (SMP) of ~−70 KPa was reached. The flag leaf and panicle tissues were collected upon attainment of SMP ~−70 kPa stress, which coincided with 50% anthesis. Plots maintained at <−10 kPa SMP served as control. The flag leaf relative water content (RWC) was measured following Barrs and Weatherley (1962) [52]. Soil moisture content (SMC) was measured by gravimetric method.

### 3.2. Identification of PTK Gene Family Members in Rice

To identify members of rice *PTK* gene family, a keyword search was performed using “Tyrosine” in the “Putative Function Search Tool” in Rice Genome Annotation Project Release 7 (RGAP, http://rice.plantbiology.msu.edu/). Twenty seven genes annotated as “tyrosine protein kinase domain containing protein” were identified from the rice genome. The nucleotide sequences and protein sequences were downloaded for further use. Protein sequences of the identified PTK homologs were subjected to domain analysis using PFAM (https://pfam.xfam.org/) and HMMER scan (https://www.ebi.ac.uk/Tools/hmmer/search/hmmscan). The sequence logos were generated using web logo tool (https://weblogo.berkeley.edu/logo.cgi).

The physical location of all identified *PTK* genes on chromosome was mapped using Chromosome Map Tool (http://viewer.shigen.info/oryzavw/maptool/MapTool.do). Gene structure of *PTK* gene, showing their exon-intron boundaries and inter phase, was generated using GSDS server (http://gsds.cbi.pku.edu.cn/).

The protein sequences of putative OsPTKs were used for BLAST search of *Arabidopsis thaliana* genome at Ensembl Plants (http://plants.ensembl.org/index.html) and homologs were identified. After removing the redundant genes, a domain analysis was carried out. The protein sequences of rice and *Arabidopsis* were used for phylogenetic analysis using Molecular Evolutionary Genetics Analysis software version 7.0 (MEGA7) [53]. The amino acid sequences were aligned using the Clastalw program and Neighbor joining analysis was performed with the pairwise deletion option with Poisson correction, bootstrap analysis was conducted with 1000 replicates. Subcellular localization of PTKs was predicted by using TargetP 1.1 [54] and LOCALIZER 1.0 [55].

### 3.3. Promoter Analysis

For in silico analysis of conserved *cis* elements in promoters of *PTK* genes, 2 kb sequences upstream of translational start codon were analyzed using PLACE database (https://www.dna.affrc.go.jp/PLACE/)

### 3.4. In Silico Analysis of PTK Expression under Abiotic Stresses

The in silico gene expression analysis was done using GENEVESTIGATOR software [56] (https://genevestigator.com/gv/index.jsp). The locus IDs of all the PTK genes were given as input in the development and perturbations tool and searched against experiments involving seedling root, shoot, panicle and flag leaf under drought and cold stresses. Expression potential represents the normalized expression value for a gene across all experiments available in the database [56] (https://genevestigator.com/gv/index.jsp).

For analysis of Abscisic acid and Jasmonic acid mediated regulation of PTK genes, Rice Expression Profile Database (RiceXPro) version 3.0 (https://ricexpro.dna.affrc.go.jp/) [57] was used. In this data base, the data sets RXP_1001 (Root gene expression profile in response to abscisic acid), RXP_1006 (Shoot gene expression profile in response to abscisic acid), RXP_1011 (Root gene expression profile in response to jasmonic acid) and RXP_1012 (Shoot gene expression profile in response to jasmonic acid) were used. These data sets were generated by microarray analysis of 7-days old seedlings of japonica rice cultivar Nipponbare treated with Abscisic acid (50 μM) or Jasmonic acid (100 μM) in Yoshida’s nutrient solution under hydroponic condition in a growth chamber at 28 °C under continuous light. mRNA isolated from different time interval were labeled with Cy3 (control) and Cy5 (hormone treatment) for microarray hybridization.

### 3.5. RNA Isolation, cDNA Synthesis, and Quantitative RT-PCR

Total RNA from seedling root and shoot, flag leaf and panicle was isolated using TRIzol reagent (Invitrogen). The quantitative and qualitative analysis carried out using spectrophotometer (Nano Drop Nd-1000 UV/Vis Spectrophotometer) and 1.2% agarose gel, respectively. Of the total RNA treated with DNase I (TURBO DNA-free ™ Kit (Ambion), 4.5 μg was reverse transcribed using Superscript III Reverse Transcription kit as per manufacturer’s instructions (Invtrogen). The sequence of primers used for qRT-PCR analysis as provided in Appendix A. Quantitative PCR (qPCR) was performed with three biological and three technical replicates with appropriate primers using PoweUpSYBR^®^green Master Mix as per manufacturer’s instructions (Thermo Fisher, USA). qRT-PCR was carried out in StepOne real time PCR machine (Applied Biosystems). Expression data were normalized using endogenous control gene *OsUBIQUITIN5* (*UBQ5*). Relative fold change was calculated using the 2^−ΔΔCt^ method [58]. The expression was represented in the form of relative fold change in expression of genes under stress conditions as compared with its expression under control conditions. The R package software (R studio) was used for analysis of variance for gene expression data and different letters denote significant variation (*p* < 0.05).

## 4. Conclusions

Analysis of 27 annotated putative PTKs in rice genome led to the identification 18 PTKs with protein tyrosine kinase motif. The kinase domains of PTKs showed high homology with that of *Arabidopsis* BRI1, a well characterized dual specific kinase, suggesting that the 18 PTKs identified in this study are dual specific kinases. We identified *Arabidopsis* homologs of rice PTKs and showed that most of the PTKs are homologous to RLCK VII group of genes from *Arabidopsis*. mRNAseq analysis using Genevestigator and qRT-PCR analysis revealed developmental and abiotic stress responsive expression pattern of *PTKs* in rice. Expression analysis suggested that most of the *PTKs* are downregulated by drought stress, but were upregulated under temperature (heat and cold), and submergence stresses. The upregulation of transcript levels under abiotic stresses mostly varied from 1.5 to 3 folds. Since regulatory function of kinases occurs through phosphorylation, even small increase in transcript levels might be sufficient for regulation stress response. Furthermore, RLCKs in association with receptor kinases (RKs) have been shown to play important role in signaling, and several plant processes and response to abiotic and biotic stresses. This study identified *PTK2*, *PTK8*, *PTK13*, *PTK14*, and *PTK18* as multiple abiotic stress (osmotic, heat, cold and submergence) upregulated genes. *PTK14* was preferentially upregulated in cold and submergence tolerant rice genotypes but was down regulated in susceptible genotypes. These *PTKs* may contribute to multiple stress tolerance in rice. The abiotic stress regulated *OsPTKs* identified in this study lays foundation for function validation of this enigmatic family of tyrosine kinases by using transgenic overexpression and CRISPR-Cas genome editing approaches.

## Figures and Tables

**Figure 1 plants-09-00664-f001:**
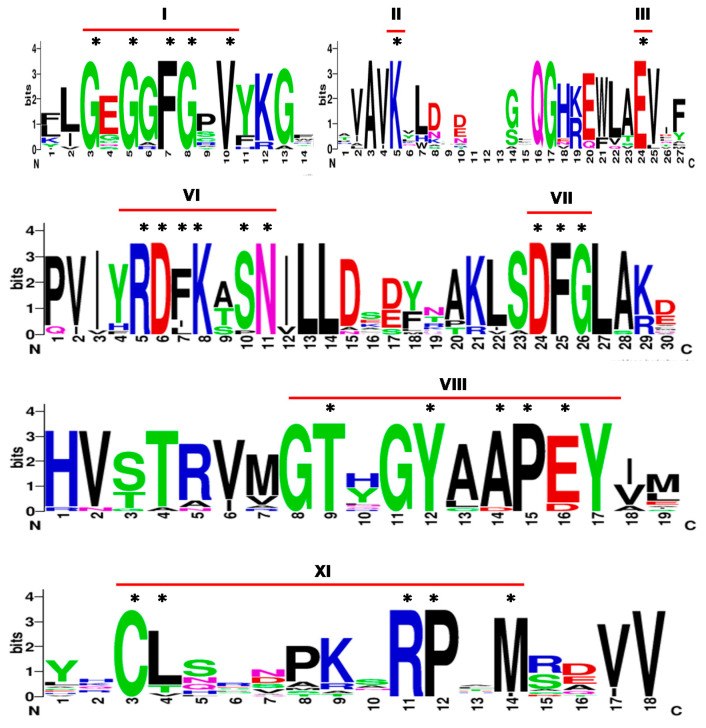
Sequence logo of conserved motifs in protein tyrosine kinases of rice. The sequence logos are based on multiple alignment analysis of 18 rice PTK proteins. The bit score indicates the residue content for each position in the sequence. * conserved residues in each sub-domain. Roman numbers I-XI denote sub-domains in kinase domain.

**Figure 2 plants-09-00664-f002:**
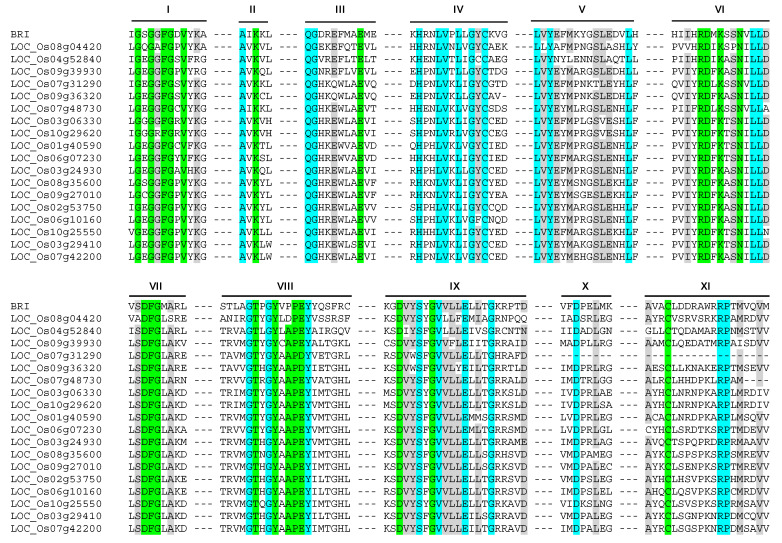
Multiple sequence alignment of rice PTK proteins with BRI (BRASSINOSTEROID INSENSITIVE 1). Roman numbers denote sub-domains of protein kinase. The codes for shades are: Green, conserved and essential for kinase activity; Cyan, conserved; Grey, >80% conservation with BRI.

**Figure 3 plants-09-00664-f003:**
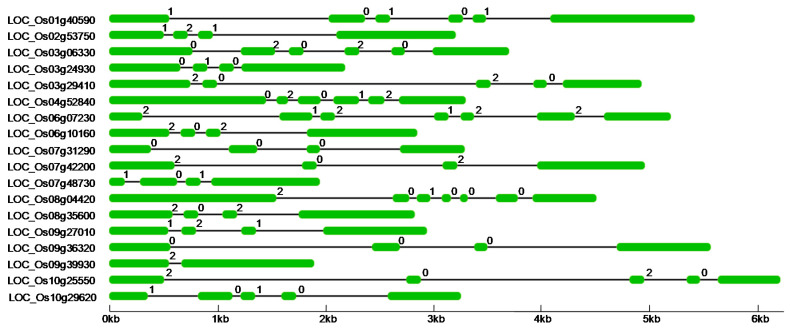
Gene structure of *PTKs* genes in rice. Coding sequence (CDS) is shown as green color box, intron as Block line and 0, 1, 2 on the introns depics intron phase.

**Figure 4 plants-09-00664-f004:**
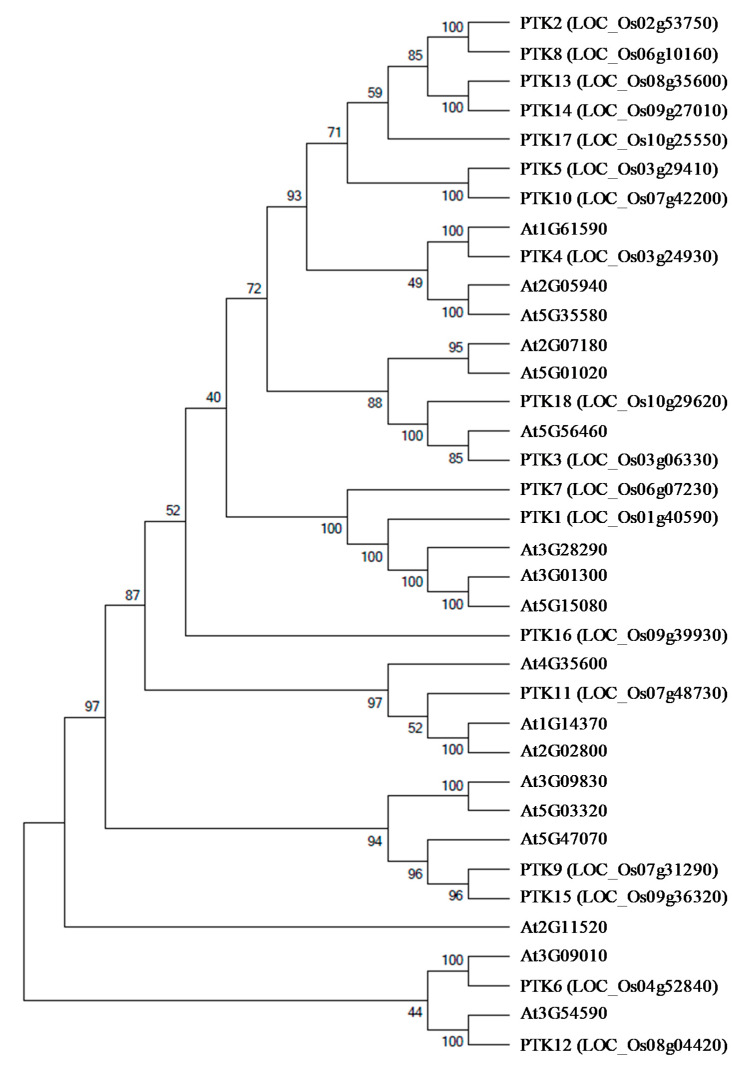
Phylogenetic analysis of OsPTK and AtPTK protein sequences. The unrooted phylogenetic tree was constructed using Molecular Evolutionary Genetics Analysis software version 7.0 (MEGA7). The amino acid sequences were aligned using the Clastalw and Neighbor joining analysis was performed with the pairwise deletion option with Poisson correction, bootstrap analysis was conducted with 1000 replicates. The digits on branches indicate boot strap values.

**Figure 5 plants-09-00664-f005:**
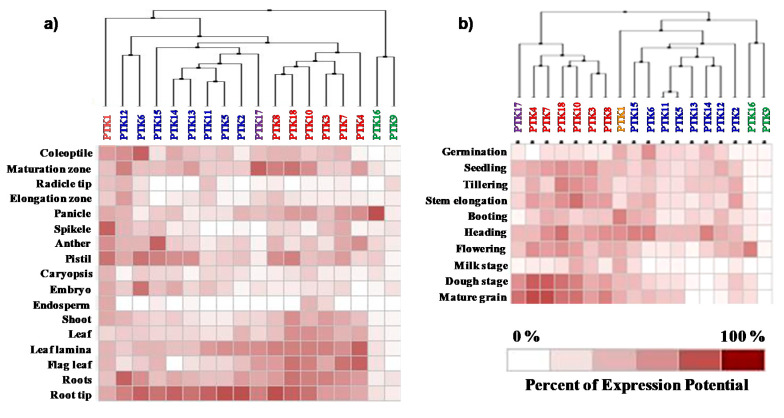
Expression analysis of *PTK* genes of rice. (**a**) Different tissues; (**b**) developmental stages. Expression analysis was carried out with mRNAseq datasets using Genevestigator. Genes in same cluster are given same color to facilitate easy comparison between (**a**) tissues and (**b**) developmental stages.

**Figure 6 plants-09-00664-f006:**
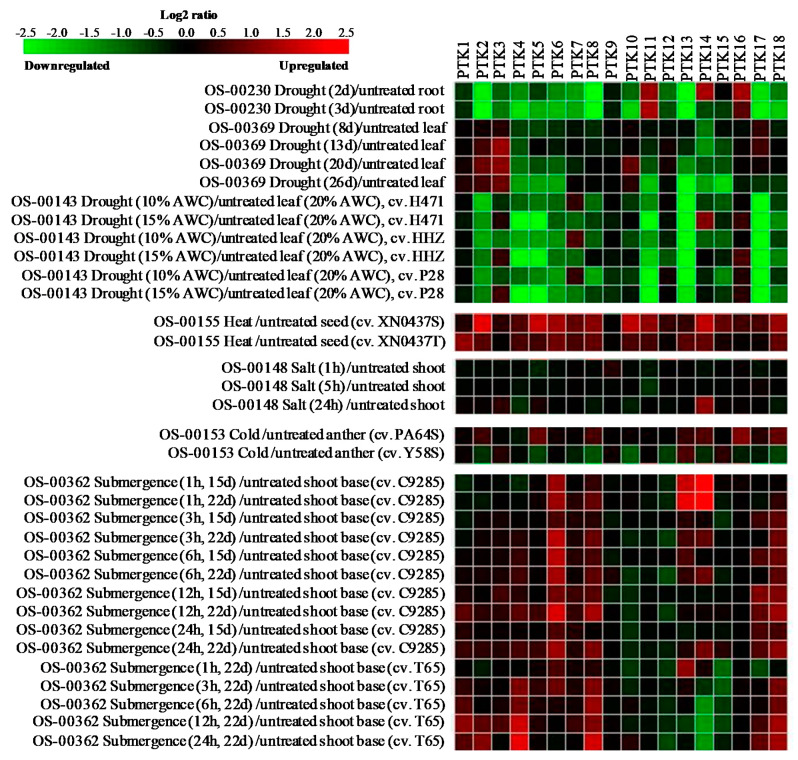
Expression analysis of *PTK* genes under different abiotic stress conditions in rice. Expression analysis was carried out with mRNAseq datasets using Genevestigator.

**Figure 7 plants-09-00664-f007:**
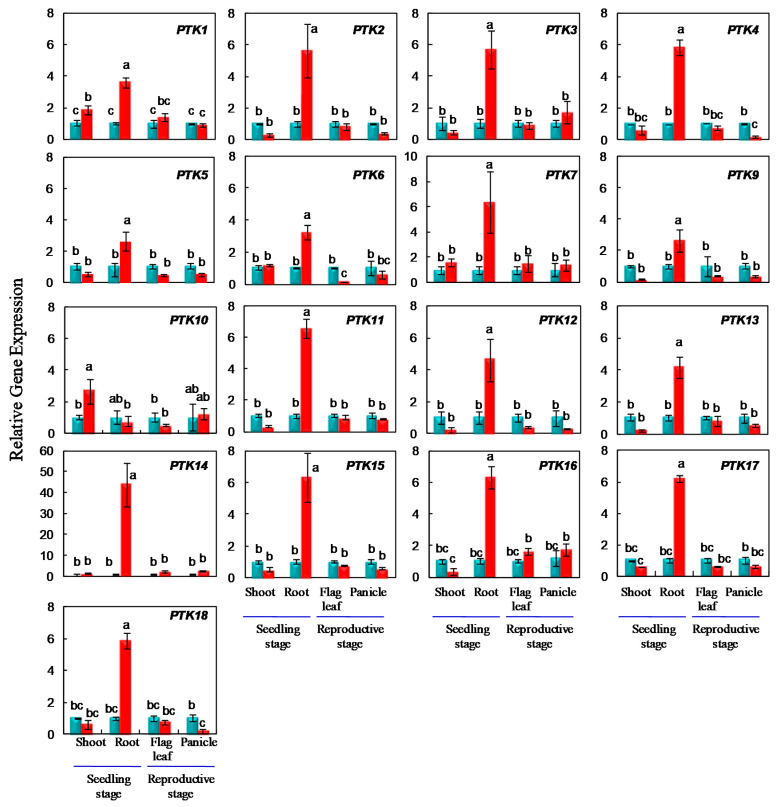
Expression analysis of *PTK* genes under osmotic stress at seedling stage in hydroponics, and under drought stress at anthesis stage in field conditions in rice. The relative expression was analysed by 2^−ΔΔct^ method. Ct values of control served as calibrator. Bars in blue and red colors represent control and stress treatments, respectively. Small letters a, b, and c represents statistical difference, different letters indicate significant differences; Error bars indicate SE; *n* = 3 biological replicates.

**Figure 8 plants-09-00664-f008:**
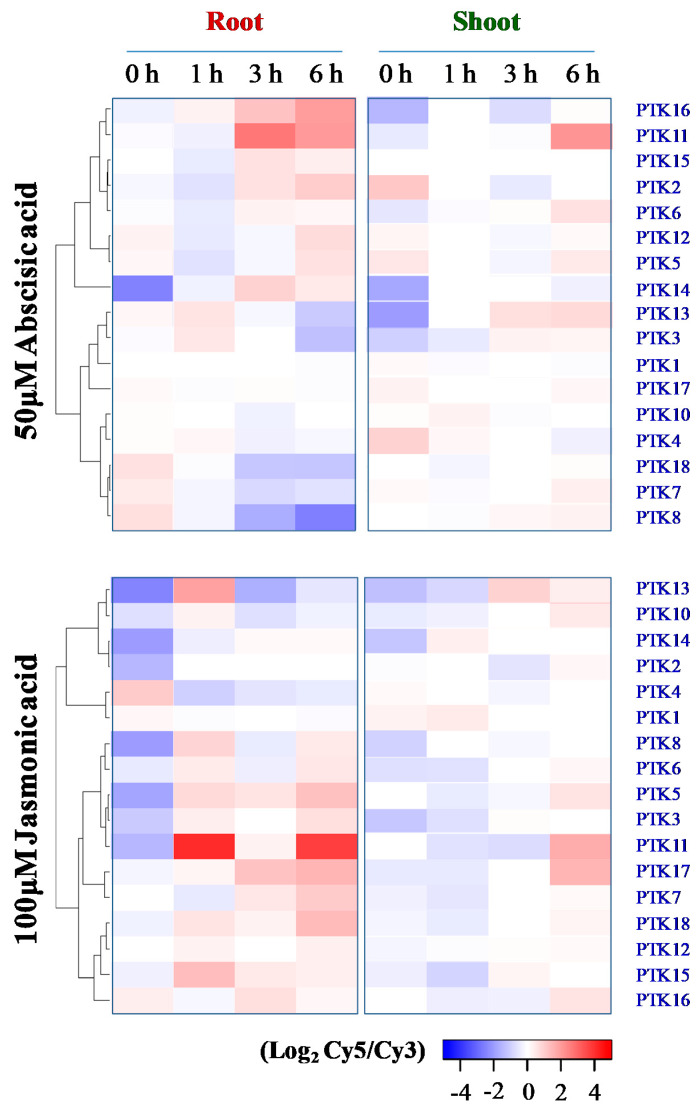
Expression analysis of *PTK* genes under ABA and Jasmonic acid treatment in rice. Expression analysis was carried out with Rice Expression Profile Database (RiceXPro) version 3.0 (https://ricexpro.dna.affrc.go.jp/). Seven days old seedlings of japonica rice cv. Nipponbare was treated with Abscisic acid (50 μM) or Jasmonic acid (100 μM) in Yoshida’s nutrient solution. Cy3 (control) and Cy5 (hormone treatment).

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
