# Peer review of "Characterization of Atypical Protein Tyrosine Kinase (PTK) Genes and Their Role in Abiotic Stress Response in Rice"

_plants, 2020, doi:10.3390/plants9050664_

Round 1

Reviewer 1 Report

This manuscript reports the characterization of protein tyrosine kinases (PTKs) in rice genome. For this purpose, the authors carried out an in silico analysis of annotated PTK genes in the rice genome and their expression analysis under osmotic and drought stress at seedling and reproductive stage, respectively. Domain analysis of the 27 annotated PTK genes in the rice genome led to the identification of 18 PTKs with tyrosine kinase domains. The authors conclude that most of the PTKs are orthologous to RLCK VII group of genes from Arabidopsis being most PTKs downregulated by drought and salt stress, but upregulated by heat, cold, and submergence stresses.

In the opinion of this reviewer, the message is clear, the paper is well-written, though there are some typos and inaccuracies, and the experimental design is fine. This work may path the way for further studies on plant PTK functions especially in rice. Notwithstanding, I have some major and minor concerns that should be addressed for the manuscript to be suitable for publication:

Major concerns

I think that some statements made in section 2.3 (Real-time RT-PCR analysis of expression of PTK genes) should be carefully revised and accordingly modified (see below).

- Lines 240-241. It can be read: “PTK7, PTK14, and PTK 16 were moderately upregulated in both flag leaf and panicle under drought stress and the upregulation varied from 1.5 to 2.7 folds (Figure 7).” However, the increased expression of these genes due to drought stress was not statistically significant for any of them. Hence, it can´t be stated that they were upregulated not even moderately.

- Lines 242-243. It can be read: “PTK3 was upregulated by 1.7 fold in panicle while it was down regulated in flag leaf. Remaining PTKs were down regulated under drought stress in both flag leaf and panicle.” I do not agree with these statements. First, up- and down-regulation of PTK3 in panicle and flag leaf were not statistically significant according to Figure 7. If the authors count these differences as relevant, PTK7, PTK10 and PTK16 also displayed increased (but not significant) expression in response to drought stress in reproductive stage. Second, I do not understand why changes in transcript levels of PTK4, PTK6, PTK9 and PTK12 are not mentioned. For these genes, a significant downregulation was found at least at one of the two stages tested. Instead, they focused on PTK7, PTK14 and PTK16 with no statistically significant differences in expression.

- Lines 248, 249, 256 and 257. In these lines it can be read that Arabidopsis gene At1G61590 (SCHENGEN1) is an ortholog of OsPTK4, but also that At2G05940 (RIPK/ACIK1A/PBL14) and At5G35580 (AvrPphB SUSCEPTIBLE1-LIKE13, PBL13), are the orthologs of OsPTK4. This must be clarified. 

- Lines 328-329. It can be read: “Expression analysis suggested that most of the PTKs are downregulated by drought and salt stress,”. I do not agree with this statement because according to Fig. 6, the expression of most of the PTK genes (except PTK14) was unaltered in response to salt stress. My conclusion is in line with the statement made in lines 193 and 194: “Expression levels of PTKs were largely unaltered under salt stress except PTK14 which showed about 1.5 fold increase at 24 h of salt stress (OS-00148; Figure 6)”. I suggest remove “salt stress” from line 329 of the Discussion.

- The Statistical test used to analyse differences in gene expression by qRT-PCR is not reported. This must be clearly described.  

- It would have been worth to study by qRT-PCR the expression of PTK genes in response to salt stress. At least those PTKs most responsive to osmotic or drought stress.

- I miss in Figure 6 the in silico analysis of the response of the PTK genes to the phytohormone abscisic acid (ABA) given that ABA plays a fundamental role in plant responses and adaptation to abiotic stress. Hence, it would be worth to show the expression of the PTK genes in rice plants exposed to ABA.

Minor concerns

- Figure 3. Resolution and size of letters should be increased. In the legend, replace “gens” with “genes”.

- Figure 4. Indicate in the legend that the phylogenetic tree is “unrooted”.

- Lines 170-171: “At milk stage of grain development, except PTK1, PTK3 and PTK10, others were down regulated.” I suggest to rewrite this sentence to make it clearer. It is a bit confusing.

- Lines 172-173: “Interestingly, during grain development (milk, dough and mature grain) stages, PTK12, PTK13 and PTK14 were downregulated (Figure 5b)”. However, I suggest also mention the high expression of PTK17, PTK4, PTK7, PTK18 and PTK10 in grain development which suggests a role for these genes in this process.

- Figure 4. Explain the use of different colors for the PTK genes.

- Line 201. Replace “while these genes downregulated by 1.5 to 2 fold in Y58S” with “while these genes were downregulated by 1.5 to 2 fold in Y58S”.

- Line 202. “in both the genotypes” should be “in both genotypes”.

- Figure 7. Panel showing the expression of the PTK11 gene. I wonder if the difference in gene expression among shoot and root at seedling stage was statistically significant, given that this difference looks like the same as those among roots and shoots for PTK12 and PTK13 and these two were significant.

- Figure 7. Explain the use of blue and red colors in the legend of the figure.  

- Line 224. Replace “highest (46.75 folds) fold induction for PTK 14 gene under osmotic stress (Figure 7)” with “highest (46.75 folds) induction for PTK14 gene under osmotic stress (Figure 7)”.

- Line 295. Replace “and” with “a”.

Reviewer 2 Report

Authors investigated protein tyrosine kinases (PTK) in rice mostly using in silico methods. They presented only one own measurement in the main text. At the same time, the role of PTKs is well documented in other plant species, which is confirmed the many accessible data in Genvestigator, respectively.

Authors should show in the introduction, why PTKs are important under drought/osmotic stress in plants. However, most of the PTK genes are downregulated under droughts stress based on the Genevestigator. From this aspect, why is important to investigate?

The only investigation of gene expression is not enough to understand the role of PTKs in cell signalling. Moreover, investigation of this crucial component after many days of stress sensing is meaningless. I think the elevated expression of PTKs in root in case of osmotic/drought stress is not surprising, because this organ senses the changes this environmental stimulus. It will be more interesting and could be a scientific question if Authors investigated gene expression in different zones of root under stress condition is time-dependent manner. Result of Genevestigator also confirmed the cell-specific changes in PTK expression.

In addition, I think 2-weeks-long PEG treatment is long to detect the potential role of PTKs in root signalling. Based on the result of Genvestigator, Other Scientists also analysed transcript levels after several hours!

It would be more interesting is a drought susceptible genotypes will be also used for the investigation and Authors could compare the two genotypes under stress condition.

To carefully describe the potential role of PTKs in stress sensing and signalling, promoter analysis could help. I think Authors should analyse the common and cell-specific CREs in PTKs promoter.

The more precise discussion of orthologes in Arabidopsis, e.g. PTK3,1,6,9,15, is needed based on Fig. 1.  

Many relevant articles are not cited e.g.

  • van Bentem, S. D. L. F., & Hirt, H. (2009). Protein tyrosine phosphorylation in plants: more abundant than expected?. Trends in plant science14(2), 71-76.
  • Luan, S. (2002). Tyrosine phosphorylation in plant cell signaling. Proceedings of the National Academy of Sciences99(18), 11567-11569.

L. 39, 63, 324,325…. Please write Arabidopsis with italic.

L.175. Fig 5 a and b: I would like to suggest the same ordering of PTKs in the two-part of the figure.

L.197. Please sign the plant species in the figure legend.

L. 217. The red and blue colour is not known. Please add a more accurate description to the figure.

Reviewer 3 Report

The Ms of Elangova et al reports on the characterization of 18 novel protein tyrosine kinase (PTK) genes from rice. The research strategy is innovative, constituting an excellent contribution to the fundamental knowledge of plant response mechanisms to abiotic stresses, whose understanding is of utmost importance in the present context of climate change, population growth and loss of agricultural land. The Ms is well written and in my opinion can be accepted for publication in MDPI Plants, provided minor revisions are included:

(1) Author names are not in the same format

(2) Moderate English editing should be done (I have pointed a few examples in the attached file)

(3) In section 2.2. the mRNA seq data are not discussed. Please include the discussion, base on the results and published data. If possible, elaborate a scheme or model of gene interactions vs stress responses vs stress tolerance/susceptibility.

(4) Highlight and discuss the different expression patterns of salt/drought stress vs temperature stress.

(5) Is the rice cultivar used in this work, tolerant to abiotic stress? please provide more information about the choice of this cultivar and bring it to the discussion.

(6) Please address the major contributions of this work in the scope of the main agricultural challenges for this century, highlighting which of these genes can be used for marker-assisted selection.

Reviewer 4 Report

This manuscript characterized rice genes for protein tyrosine kinase by in silico analysis and examined their tissue-specific and stress-responsive expression. The results will help understand their role in rice physiological phenomenon. I have some concerns to be addressed or corrected before publication as below.

You identified 18 PTK genes in rice and mentioned that the kinase domain shared high homology with that of BRI1. I wonder if the 18 PTK genes include rice BRI1 ortholog? Please give a comment on this point.

Lines 80-82. Do you think that LOC_Os02g39560 lacking subdomain I is pseudogene?

Line 90. “VIB2 may be wrong. “VI” may be correct.

Fig. 1. Please show what roman numbers, I, II, … XI, indicate in figure legend.

Line 108. “implicating that these two are dual-specificity kinases”. It is better to experimentally verify phosphorylation activity of some PTKs and the target amino acids.

Line 134. pI, isoelectric point

Line 139. Please add explanations for two words, “cTP” and “mTP”.

Fig. 7. What do red and blue bars indicate? Besides, how did you determine “significant differences”? Which statistical test was used? p- or q-values?

Line 260. Delete “(Lin et al. 2015)”.

Round 2

Reviewer 1 Report

In this revised version, the authors have satisfactorily answered most of the questions raised in my previous review which has improved the manuscript. Nevertheless, I still have some concerns and comments.

Line 296. A reference for this statement is missing: “As ABA-dependent signaling plays a major role in stress inducible gene expression”.

Lines 298-299. I suggest changing the sentence: “ABA treatment upregulated PTK3 and PTK13 at 1h, but were downregulated at 6h in roots” with this: “ABA treatment moderately upregulated PTK3 and PTK13 at 1h, but were downregulated at 6h in roots”.

Lines 299-301. I suggest changing the sentence: “PTK2, PTK11, PTK14, PTK15 and PTK16 were upregulated by ABA at 3h, while PTK5 and PTK12 were also upregulated at 6h in roots of rice seedlings” with this: “PTK2, PTK11, PTK14, PTK15 and PTK16 were upregulated by ABA at 3-6h, especially PTK11, while PTK5 and PTK12 were upregulated at 6h in roots of rice seedlings”

Line 302-303. I suggest changing: “In the shoots of rice seedlings, PTK11 was highly upregulated, while PTK5, PTK6 and PTK13 were moderately upregulated by ABA” with this: “In the shoots of rice seedlings, PTK11 was highly upregulated at 6h of ABA treatment, while PTK5, PTK6 and PTK13 were moderately upregulated”.

Lines 309-314. I recommend to rewrite the JA paragraph. It is confusing. I suggest replacing the current paragraph:

“PTK5, PTK8, PTK11, PTK13, PTK15 were upregulated at 1 h in roots by JA. By 3 or 6h of the treatment, PTK7, PTK17 and PTK18 were also upregulated in roots by JA. Interestingly of these upregulated PTKs, PTK13 and PTK15 were downregulated at 6h. PTK4 was downregulated by JA in roots of rice seedlings (Figure 8). PTK11, PTK13 and PTK17 were also upregulated in shots of rice seedlings by JA (Figure 8). Thus, PTK5 was upregulated, and PTK11 and PTK13 were upregulated both in roots and shoots by both ABA and JA (Figure 8)”

with this one:

“PTK5, PTK8, PTK11, PTK13, PTK15 were upregulated at 1 h in roots by JA, but PTK13 and PTK15 were downregulated at 3-6 h. By 3 or 6h of treatment, PTK3, PTK5, PTK7, PTK11, PTK17 and PTK18 were also upregulated in roots by JA (Figure 8). As regards shoots, PTK11, PTK13 and PTK17 were upregulated by JA at 3 and/or 6 h of treatment (Figure 8). Interestingly, PTK11 and to a lesser extent, PTK5 and PTK13, were upregulated in roots and shoots by both ABA and JA (Figure 8)”.

Lines 429-430. Replace “different alphabets” with “different letters”.

Author Response

We thank the reviewer for critical review and constructive suggestions which improved our manuscript.

All the corrections suggested have been incorporated in the revised version. 

Reviewer 2 Report

Thank you, Authors carefully improved their manuscript based on my comments.

Author Response

We thank the reviewer for critical reviewing and improving our manuscript.